# The Role of GeneXpert^®^ for Tuberculosis Diagnostics in Brazil: An Examination from a Historical and Epidemiological Perspective

**DOI:** 10.3390/tropicalmed8110483

**Published:** 2023-10-26

**Authors:** Tirça Naiara da Silva Iúdice, Marília Lima da Conceição, Artemir Coelho de Brito, Nicole Menezes de Souza, Cristal Ribeiro Mesquita, Ricardo José de Paula Souza e Guimarães, Ismari Perini Furlaneto, Alessandra de Souza Saboia, Maria Cristina da Silva Lourenço, Karla Valéria Batista Lima, Emilyn Costa Conceição

**Affiliations:** 1Programa de Pós-Graduação Biologia Parasitária na Amazônia, Universidade do Estado do Pará, Belém-Pará 66087-670, PA, Brazil; tircaiudice@gmail.com (T.N.d.S.I.); cristalmesquita@yahoo.com.br (C.R.M.); 2Seção de Bacteriologia e Micologia, Instituto Evandro Chagas, Ananindeua 67030-000, PA, Brazil; marilimadc@msn.com; 3Coordenação-Geral de Vigilância da Tuberculose, Micoses Endêmicas e Micobactérias Não Tuberculosas—CGTM, Ministério da Saúde, Brasília 70000-000, DF, Brazil; artemir.brito@saude.gov.br (A.C.d.B.);; 4Laboratório de Geoprocessamento, Instituto Evandro Chagas, Ananindeua 67030-000, PA, Brazil; ricardojpsg@gmail.com; 5Curso de Medicina, Centro Universitário do Estado do Pará, Belém 66087-662, PA, Brazil; ismaripf@hotmail.com; 6Laboratório de Bacteriologia e Bioensaios em Micobactérias, Instituto Nacional de Infectologia Evandro Chagas, Fundação Oswaldo Cruz, Manguinhos, Rio de Janeiro 21040-360, RJ, Brazil; allessandrasaboia@gmail.com (A.d.S.S.); cristina.lourenco@ini.fiocruz.br (M.C.d.S.L.); 7Pós-Graduação em Pesquisa Clínica e Doenças Infecciosas, Instituto Nacional de Infectologia Evandro Chagas, Fundação Oswaldo Cruz, Manguinhos, Rio de Janeiro 67030-000, RJ, Brazil; 8Department of Science and Innovation–National Research Foundation Centre of Excellence for Biomedical Tuberculosis Research, South African Medical Research Council Centre for Tuberculosis Research, Division of Molecular Biology and Human Genetics, Faculty of Medicine and Health Sciences, Stellenbosch University, Cape Town 7505, South Africa

**Keywords:** tuberculosis, rapid molecular test, Xpert MTB/RIF, Xpert MTB/RIF Ultra, Brazil

## Abstract

The rapid molecular test (RMT) performed on the GeneXpert^®^ system is widely used as a control strategy and surveillance technique for tuberculosis (TB). In the region of the Americas, TB incidence is slowly increasing owing to an upward trend in Brazil, which is among the high TB-burden countries (HBCs), ranking in the 19th position. In this context, we aimed to (i) describe the implementation and history of RMT-TB (Xpert^®^ MTB/RIF and Xpert^®^ MTB/RIF Ultra) in Brazil; (ii) to evaluate the national RMT laboratory distribution, TB, and resistance to RIF detection by RMT; and (iii) to correlate these data with Brazilian TB incidence. The quantitative data of Xpert^®^ MTB/RIF and Xpert^®^ MTB/RIF Ultra assays performed in the pulmonary TB investigation from 2014 to 2020 were provided by the Brazilian Ministry of Health. A spatial visualization using ArcGIS software was performed. The Southeast region constituted about half of the RMT laboratories—from 39.4% to 45.9% of the total value over the five regions. Regarding the federal units, the São Paulo state alone represented from 20.2% to 34.1% (5.0 to 8.5 times the value) of RMT laboratories over the years observed. There were significant differences (*p* < 0.0001) in the frequency of RMT laboratories between all years of the historical series. There was an unequal distribution of RMT laboratories between Brazilian regions and federal units. This alerts us for the surveillance of rapid molecular detection of TB in different parts of the country, with the possibility of improving the distribution of tests in areas of higher incidence in order to achieve the level of disease control recommended by national and worldwide authorities.

## 1. Introduction

Tuberculosis (TB) continues to represent a significant global health concern, especially in countries with lower and middle income levels. Addressing the complexities of TB in these regions demands a multidisciplinary strategy that integrates expertise from diverse fields, including medicine, public health, epidemiology, and social sciences. A fundamental obstacle in the battle against TB within these countries pertains to the insufficient supply of suitable diagnostic methodologies [1].

On the South American continent, there has been an observed gradual escalation in the incidence of TB, primarily attributed to an upward trajectory observed in Brazil, which is ranked 19th among countries with a high TB burden (HBCs) [2]. Within Brazil, variations in the TB burden among different states are elucidated by the concentration of TB transmission in peripheral urban areas, specifically in clusters of patients residing in overcrowded and substandard living conditions within certain cities. These disparities are further compounded by social determinants associated with the susceptibility of individuals affected by TB [3].

In order to effectively combat the TB epidemic, the World Health Organization (WHO) has executed the End TB Strategy, with the overarching objectives of reducing TB-related mortality by 95% and curbing the incidence of new cases by 90% by the year 2035. To successfully realize these objectives, it is imperative to ensure the timely and precise diagnosis of all TB manifestations and the rapid detection of drug resistance (DR) [4,5,6].

In this context, with regard to the requirements of the healthcare system, it is imperative to enhance access to swift TB diagnosis and drug susceptibility testing (DST) through the expansion and fortification of the diagnostic service network. This expansion should encompass the adoption of WHO-recommended rapid molecular tests (RMTs) for initial diagnostic evaluations, with a particular focus on drug resistance detection. Notable examples of such RMTs include the Xpert^®^ MTB/RIF Ultra (Cepheid, Sunnyvale, CA, USA) and the Truenat^®^ MTB-RIF Dx system (Molbio Diagnostics Pvt. Ltd., Verna, Goa, India). These technologies should be strategically integrated into peripheral healthcare services in order to facilitate early and precise TB diagnosis, with an emphasis on identifying rifampicin (RIF) resistance, especially among high-risk groups [4,5,6].

The suitability of RMTs for deployment as a TB screening tool in community-wide systematic screening initiatives has been reviewed, particularly in regions where TB prevalence stands at 0.5% or higher as well as among populations at elevated risk of TB [7]. The Xpert^®^ MTB/RIF Ultra has the capability to detect the presence of the *Mycobacterium tuberculosis* complex (MTBC) in patient specimens, along with the primary mutations responsible for RIF resistance, within a remarkably short timeframe. Specifically, it can provide results within 77 min for positive cases and 66 min for negative cases, with an average processing time of 69 min [8].

In this context, the intensified use of RMT for TB diagnosis has a substantial influence on TB surveillance and the impact of case detection. Thus, our objective was to evaluate the utilization of the GeneXpert^®^ system assay (Cepheid, Sunnyvale, CA, USA) in Brazil from both a historical and epidemiological standpoint.

## 2. Materials and Methods

### 2.1. Study Design

This is a cross-sectional epidemiological study that analyzed secondary data sourced from the Brazilian Health Ministry. The primary objectives of this study were as follows: (i) to provide a comprehensive overview of the introduction and historical context surrounding the utilization of RMT-TB, specifically Xpert^®^ MTB/RIF and Xpert^®^ MTB/RIF Ultra, in Brazil; (ii) to assess the nationwide distribution of RMT laboratories and the capacity for TB and rifampicin (RIF) detection through RMT-TB equipment, delineated by year, federal units, and regions; and (iii) to establish correlations between these data and the incidence of TB in Brazil.

### 2.2. Setting

We divided the geographical map of Brazil into its five primary regions (North—N, Northeast—NE, Southeast—SE, Central West—CW, and South—S) and further segmented it into 27 individual federal units (states). This stratification was employed in order to facilitate the epidemiological analysis and historical examination of the GeneXpert^®^ system equipment’s deployment in Brazil (Figure 1).

### 2.3. GeneXpert MTB/RIF Equipment Data (RMT-TB Equipment)

The GeneXpert^®^ system comprises both the equipment system and the individual assay cartridges, specifically Xpert MTB/RIF and Xpert MTB/RIF Ultra. This technology was introduced into the Brazilian Unified Health System (Sistema Único de Saúde—SUS) in 2014 utilizing the Xpert MTB/RIF cartridge and was subsequently upgraded to the Xpert MTB/RIF Ultra version in 2020 [9,10].

### 2.4. Data Collection of the Historical Context

To delve into the historical context of the GeneXpert^®^ system technology in Brazil, our research spanned from August to September 2021, involving an examination of official Brazilian documents that provided guidance for the integration of this diagnostic tool into the SUS under the purview of the Brazilian Ministry of Health [11]. We sourced these documents from three websites: (1) the Biblioteca Virtual em Saúde do Ministério da Saúde (https://bvsms.saude.gov.br/ accessed on 16 April 2021); (2) the Diário Oficial da União (https://www.gov.br/imprensanacional/pt-br accessed on 16 April 2021); and (3) the Ministry of Health (https://www.gov.br/saude/pt-br accessed on 16 April 2021).

For our analysis, we explored the following key topics: (i) the regulatory framework governing GeneXpert^®^ system equipment in Brazil; (ii) the interest exhibited by the Brazilian Ministry of Health in the adoption of GeneXpert^®^ system equipment within the SUS; (iii) the procedural aspects pertaining to the implementation of the GeneXpert^®^ system; (iv) the authorization procedures for the utilization of the GeneXpert^®^ system; and (v) the commencement of operations involving RMT-TB equipment within the SUS.

### 2.5. GeneXpert^®^ System Quantitative Data Collection

We acquired quantitative data pertaining to the results of Xpert^®^ MTB/RIF and Xpert^®^ MTB/RIF Ultra assays for pulmonary TB cases diagnosed from 2014 to 2020 through collaboration with the General Coordination for the Monitoring of Chronic Respiratory Transmission Diseases (Coordenação-Geral de Vigilância das Doenças de Transmissão Respiratória de Condições Crônicas—CGDR) at the Brazilian Ministry of Health, now called Coordenação-Geral de Vigilância da Tuberculose, Micoses Endêmicas e Micobactérias Não Tuberculosas (CGTM), currently reformulated as the General Coordination for Surveillance of Tuberculosis, Endemic Mycoses and Non-Tuberculous Mycobacteria (Coordenação-Geral de Vigilância da Tuberculose, Micoses Endêmicas e Micobactérias Não Tuberculosas—CGTM). These data were obtained via the Electronic Citizen Information System (Sistema Eletrônico de Informações ao Cidadão-e-SIC).

### 2.6. Health Care Indicators

We selected healthcare indicators based on the criteria outlined by the Pan American Health Organization, including (i) the absolute and relative frequency of RMT laboratories and the number of TB cases and TB cases with RIF resistance detected by the GeneXpert^®^ system within different regions and federal units of the country; (ii) the ratio of RMT laboratories per 106 inhabitants in various regions and federal units; (iii) the absolute and relative frequency of positive *Mycobacterium tuberculosis* (MTB) cases detected by the GeneXpert^®^ system among new cases of TB; and (iv) the incidence of TB cases in different regions and federal units.

The calculation of the absolute and relative frequencies of RMT laboratories and cases of MTB positivity and RIF resistance detected by the GeneXpert^®^ system was based on data provided by CGTM. The ratio of RMT laboratories per 106 inhabitants was computed using data from CGTM and population data from the Brazilian Institute of Geography and Statistics (Instituto Brasileiro de Geografia e Estatística—IBGE). The absolute and relative frequencies of positive MTB cases detected by the GeneXpert^®^ system among new TB cases were determined using data from CGTM and information collected from the National System of Information on Notifiable Diseases (Sistema de Informação de Agravos de Notificação—SINAN), available on the Department of Informatics of the Unified Health System’s website (Departamento de Informática do Sistema Único de Saúde—DATASUS) as of 27 July 2021.

The TB incidence coefficient was calculated using reports from the TB database (http://tabnet.datasus.gov.br/ accessed on 4 June 2021). Therefore, in order to calculate the TB incidence coefficient of the regions and federal units, we used the Tab for Windows—TabWin—program to apply the formula used by the Brazilian Ministry of Health [12,13], a number of new cases of TB per place of residence estimated population per 100,000 inhabitants. New TB cases are considered the sum of the entry types “new case”, “unknown”, and “post-death”, as recommended by the Brazilian data system. New TB cases for 2014 to 2020 were collected from the SINAN available on the website of the DATASUS (https://datasus.saude.gov.br/ accessed on 27 July 2021). The estimated population from 2014 to 2020 was from the website of the IBGE (https://www.ibge.gov.br/ accessed on 27 July 2021).

### 2.7. Geoprocessing

The boundaries of the regional divisions of Brazil (States and Major Regions) applied presently were obtained on the website of the IBGE (https://www.ibge.gov.br/ accessed on 27 July 2021). The coordinates were obtained from the centroids of the states using the tool “Polygon to Point” from ET GeoWizards (https://www.ian-ko.com/ETGeoWizards.html accessed on 4 August 2021) in the ArcGIS software (https://www.arcgis.com/ accessed on 4 August 2021). The ArcGIS software performed data processing, interpretation, visualization, and spatial analysis (http://www.arcgis.com/ accessed on 4 August 2021). TB incidence was classified into four levels according to the WHO: low cases (0–9.9—green color), medium (10–24.9—yellow), high (25–49.9 cases—orange) and very high (>50—red) cases/100,000 inhabitants. RMT-TB detected MTB cases and rifampicin resistance cases; thus, RMT laboratories were distributed in the incidence map by quartile (MTB cases detected by RMT-TB and RIF resistance cases were distributed in the incidence map by quartile). The maps were divided into seven years (2014–2020) for better visualization, limited by federal units and major regions (N: north, NE: northeast, CW: central-west, SE: southeast and S: south).

### 2.8. Statistics Analysis

The Chi-square test of adherence was used to examine the differences between RMT laboratories’ regional frequencies each year, and the Chi-square test for trend was used to assess the trend in MTB frequency or RIF resistance throughout the historical series. Furthermore, Spearman’s rank correlation test was applied to estimate the correlation between the amount of RMT laboratory equipment, TB incidence in the region and federal units, and MTB frequency by region and national units and between RMT–laboratory ratio in each region and federal unit and TB incidence per year. The Friedman test was applied to compare RMT laboratories quantitatively in 2014 vs. 2015, 2014 vs. 2016, 2014 vs. 2017, 2014 vs. 2018, 2014 vs. 2019, and 2014 vs. 2020.

## 3. Results

### 3.1. Historical Background of GeneXpert^®^ System in Brazil

The historical background of the GeneXpert^®^ system in Brazil was built based on three official documents [12,14], two published studies [15,16], and an official document from the National Health Surveillance Agency (Agência Nacional de Vigilância Sanitária —ANVISA). The GeneXpert^®^ system equipment was registered by ANVISA in 2009 to be used by the private sector [17].

### 3.2. The Interest of the Brazilian Ministry of Health in the Implantation of GeneXpert^®^ System Equipment in SUS

The Brazilian Health Ministry showed interest in the performance of the equipment between the years 2012 and 2013. To this end, it signed a partnership with the Bill and Melinda Gates Foundation (BMGF) and the Ataulpho de Paiva Foundation, as it would promote innovation in the diagnosis and treatment of TB in Brazil, in addition to the allocation of resources for the progress of the demands of the National Tuberculosis Control Program (Programa Nacional de Controle da Tuberculose—PNCT) in improving the detection and treatment of TB in Brazil [12].

### 3.3. Process for the Deployment of the GeneXpert^®^ System Equipment in SUS

The partnership gave rise to the pilot project to deploy the assay Xpert MTB/RIF realized by GeneXpert equipment to pulmonary TB diagnosis in two Brazilian cities as a substitute for direct sputum smear microscopy under routine conditions in order to assess whether it would increase the notification rate of laboratory-confirmed pulmonary TB to the national notification system and reduce the time to initiation of TB treatment, as well as to estimate the impact that the adoption of this equipment would have on the detection of cases of RIF-resistant TB (RR-TB) used in treatment [18]. In parallel to the pilot study, a cost analysis of the assay Xpert MTB/RIF and the GeneXpert^®^ system equipment for pulmonary TB diagnosis by SUS was performed [16].

During the pilot study, the implementation of a national network of GeneXpert^®^ system equipment was addressed by the PNCT. To this end, it had some priorities: (i) municipalities that reported more than 200 TB cases in 2011 and that had a laboratory with a physical and biosafety structure equivalent to the performance of the microscopy; (ii) a cutoff point of 200 new cases, because this is the operational capacity of a GeneXpert^®^ system with two modules, being able to perform 8 to 16 tests per day. In addition, they included: (i) state capitals that did not report more than 200 new cases in 2011; (ii) municipalities that are the seat of prisons or with indigenous populations that reported at least 50 cases in 2011; (iii) the Central Public Health Laboratories (Laboratório Central de Saúde Pública-LACEN) for being the coordinators of the state network of laboratories and responsible for training and quality control of the tests performed [12].

### 3.4. Authorization of the Use of GeneXpert^®^ System Equipment by SUS

The members of the National Commission for Technology Incorporation at SUS (Comissão Nacional de Incorporação de Tecnologias no SUS—CONITEC) present at the 11th ordinary meeting in plenary on 12 July 2012 unanimously recommended the assay Xpert MTB/RIF realized by GeneXpert^®^ system equipment as a test for diagnosing TB and for detecting resistance to RIF [12].

However, the following year, they conducted public consultations between 15 January and 4 February 2013, characterized as suggestion and questioning, respectively. Finally, on 7 March 2013, the final decision of CONITEC members was to recommend the incorporation of the Xpert MTB/RIF assay for TB diagnosis and detection of resistance to RIF within the scope of the PNCT and according to the criteria established for its implementation [12]. Ordinance No. 48 of 10 September 2013, made public CONITEC’s decision to incorporate the assay Xpert MTB/RIF into SUS for TB diagnosis and detection of RIF resistance [19].

### 3.5. Start of Operation of GeneXpert^®^ System Equipment in SUS

The GeneXpert^®^ system equipment was distributed to SUS in May 2014, beginning use of the Xpert MTB/RIF assay in the same month, and was monitored by the national Rapid Test Network-TB (RTN-TB) created for this purpose the following month [9].

### 3.6. Quantitative Analysis of GeneXpert^®^ System in Brazil

In the first year (2014) of the introduction of RMT-TB in SUS, there were (n = 88 laboratories) with 149 sets of RMT-TB equipment; in 2020, this number increased to (n = 203 laboratories) with 257 sets of RMT-TB equipment. The identified increase was 130.68% in the number of RMT laboratories and 72.48% in the amount of RMT-TB equipment in the evaluated period. Furthermore, Friedman’s test only had a significant difference (*p* < 0.0001) in the number of RMT laboratories between the years 2014 vs. 2018, 2014 vs. 2019, and 2014 vs. 2020.

To evaluate the distribution of RMT laboratories among the regions, we considered that there would be a 20% portion of the total RMT laboratories per region. However, it was about twice as much, from 39.41% in the Southeast region to 45.93% (Figure 2A). Regarding the federal units, the equal distribution would be about 4% of the RMT laboratories each. Still, the state of São Paulo alone had 20.20–34.09% (5.0 to 8.5 times the value) of RMT laboratories in the evaluated period (Figure 2B). There were significant differences (*p* < 0.0001) in the frequency of RMT laboratories across all years of the historical series.

The RMT laboratory ratio/population in the Brazilian regions over the evaluated period demonstrates that in the last three years, the highest ratio was obtained in the North region, which varied approximately from 60% to 80% more than in the Central West region, the region with the second highest ratio, and about 80% more compared to Brazil as a whole (Figure 2C). Calculating the federal units individually, Rondônia had the highest percentage in the last three years, surpassing Acre and Roraima (Figure 2D).

In addition, we can observe the number of RMT laboratories in some states in which the TB incidence coefficient ranged from high to very high, as in Acre (0–3 laboratories); Roraima (1–2 laboratories), Pernambuco (5–11 laboratories), and Pará (3–6 laboratories) (Figure 3).

In seven years, (n = 508,381) new cases of MTB were registered in Brazil, and RMT-TB was responsible for detecting 205,585 (40.44%) of these recent cases. The Southeast region had the highest number of new cases of MTB (n = 232,242 cases); RMT-TB detected 106,631 (45.91%) cases. Among the federal units, the state of São Paulo had the most significant number of new cases (n = 121,748 cases); the equipment identified 59,150 (48.58%) cases.

There was a higher number of tests for MTB (n = 1,117,645) by RMT-TB in the SUS in the Southeast region, where 106,631 (9.54%) were positive. Regarding the federal units, the state of São Paulo was the one that tested the most for MTB (n = 842,557 tests), obtaining 59,150 (7.02%) positives using only RMT-TB.

The state of Amazonas presented a very high incidence (>50 cases/100,000 inhabitants) in all years of the historical series and an increase in the detection of positive MTB by RMT-TB in the year 2016 (Figure 4). On the other hand, in the state of Rio de Janeiro, which also showed very high incidence (>50 cases/100,000 inhabitants) in all years of the historical series, the detection of positive MTB was higher in two years of the historical series (2017 and 2020) (Figure 4).

In states where TB incidence ranged from high to very high during the period evaluated, Roraima in 2019 had an increase in RMT-TB-positive MTB detection that coincided with highest moment of TB incidence (>50 cases/100,000 population) in that state (Figure 4). In Acre, MTB-positive detection was highest for the first time in 2017, the same year that the state first had TB incidence at the highest level (>50 cases/100,000 population) (Figure 4). In Pernambuco, MTB-positive detection experienced an increase for the first time in 2020, one year after the state showed very high TB incidence (>50 cases/100,000 population) for the second time (Figure 4), whereas in the State of Pará, in the year 2019, when the incidence of TB was very high (>50 cases/100,000 population), the detection of positive MTB in those tested by RMT-TB was lower compared to the previous year (2018) when the incidence of TB was lower (25–49.9 cases/100,000 population) (Figure 4).

RMT-TB detected RIF resistance in (n = 8209 positive MTB) all seven years. Given this scenario, we observed that in 2020, in the state of Mato Grosso do Sul in the Central West region, the detection of RIF-resistant MTB was about (20.1% to 30.0%) of the MTB detected in the state by the RMT-TB, a higher percentage than those detected in the states of Amazonas, Rio de Janeiro, and Acre, where the incidence of TB was very high (>50 cases/100,000 inhabitants) in the same year, as well as of those detected in the other federal units in the previous years in the historical series (Figure 5).

Furthermore, it is discernible that there was a notable trend in the frequency of positive MTB detection by RMT-TB in laboratories affiliated with the Brazilian Unified Health System (SUS). Specifically, there was a significant upward trend observed in the North, Southeast, South, Central West, and nationwide. In contrast, the Northeast region experienced a substantial decreasing trend (Appendix A). Additionally, this trend analysis revealed a significant decline in the frequency of RIF-resistant MTB detection by RMT-TB within SUS laboratories in the North and Southeast regions, while the Central West region observed a notable increase (Appendix A).

Considering the correlation between health indicators, we conducted Spearman’s correlation analysis. It unveiled a significant correlation (*p* = 0.0452) between the frequency of MTB and the number of RMT laboratories in the federal units, but only in the year 2014. Additionally, a significant correlation (*p* = 0.0080) was observed between the number of RMT laboratories and TB incidence by region over the historical time frame.

## 4. Discussion

Since the introduction of the Gene Xpert^®^ system in Brazil, numerous studies have been conducted to assess its cost effectiveness, compare it to traditional or other RMT methods, evaluate its impact on patient outcomes, and examine its role in the detection of TB/RR-TB in non-sputum samples or paucibacillary samples, among other aspects. In general, these studies have consistently demonstrated that the adoption of this technology has led to significant enhancements in patient care and health outcomes across Brazil [14,16,20,21,22,23,24,25,26,27,28,29,30,31,32,33,34,35,36,37,38,39,40,41,42,43,44,45,46,47,48,49,50,51].

This has empowered healthcare providers to administer timely and appropriate treatments, ultimately resulting in improved health outcomes for patients. Furthermore, the system has facilitated an increase in the reporting of laboratory-confirmed cases of pulmonary TB while reducing the time required to initiate treatment. These developments are vital in curbing the spread of the disease and improving the chances of successful treatment before it progresses. Additionally, the substitution of traditional smear microscopy with the more efficient Xpert MTB/RIF system has been linked to a noticeable decrease in treatment discontinuations and a higher rate of TB case confirmation. All of these factors work together to elevate patient outcomes by ensuring they receive timely and effective treatment for successful disease management [14,16,20,21,22,23,24,25,26,27,28,29,30,31,32,33,34,35,36,37,38,39,40,41,42,43,44,45,46,47,48,49,50,51].

In our study, we observed as a strength that SUS has expanded its network of laboratories equipped with RMT-TB technology through the Gene Xpert^®^ system. In 2014, there were 88 laboratories with 149 sets of RMT-TB equipment, and by 2020, these numbers had increased to 203 laboratories with 257 sets of RMT-TB equipment, indicating a considerable expansion of the RMT-TB network. It is important to note that the enumeration of RMT-TB network equipment by CGTM is influenced by logistical issues in equipment distribution, which can result in disparities between the donation date, equipment installation date, and production date. This variability makes maintaining an accurate equipment count challenging, necessitating continuous updates to the CGTM database (ref).

Throughout the historical series, there was marked heterogeneity in the distribution of RMT laboratories among states. São Paulo consistently exhibited a higher presence of RMT laboratories and maintained a consistent TB incidence scenario (25–49.9 cases/100,000 population). In contrast, states such as Roraima, Acre, Pernambuco, and Pará, where TB incidence ranged from high (25–49.9 cases/100,000 inhabitants) to very high (>50 cases/100,000 inhabitants) during the study period, had a lower presence of RMT laboratories.

The detection of MTB-positive cases by RMT-TB among those tested was also higher in these states in 2019 and 2020 compared to São Paulo. The state of São Paulo has had the largest Brazilian RMT-TB installation since the beginning because it also includes equipment purchased by the local locations, which demonstrates the state’s proactivity in expanding the RMT-TB network.

While our study found a significant correlation between the frequency of MTB and the number of RMT laboratories in federal units during the initial year of our historical analysis, as well as a correlation between the number of RMT laboratories and TB incidence by region throughout the study, we suggest that future investigations should conduct direct observational follow-ups to assess the impact of the operational RMT-TB laboratories on MTB detection frequency and incidence rates in states with very high TB incidence rates (>50 cases/100,000 inhabitants) during the study period.

With the introduction of any effective diagnostic tool, such as the GeneXpert^®^ system, there will be a rise in the detection of TB cases. Likewise, strategically allocating limited resources like the GeneXpert^®^ system demands that health authorities prioritize its deployment to regions with potentially higher demand or expected positive outcomes. Hence, there exists a complex relationship between the use of the GeneXpert^®^ system and the ratios of newly diagnosed TB cases.

While examining the implementation of novel diagnostic technologies, both research and policies are actively involved, indicating potential for enhancement. In this context, we recommend an expansion of the RMT-TB network in the North and Northeast regions of Brazil. This expansion should incorporate upgrades in equipment infrastructure, technical assistance, and maintenance capabilities, particularly considering the introduction of new anti-TB medications and foreseeing the forthcoming integration of the Xpert MTB/XDR technology, which can supplement the diagnosis of DR-TB.

In addition to enhancing TB diagnosis, the initial test conducted by the GeneXpert^®^ system underwent an update, resulting in the Xpert MTB/RIF test, which was later rebranded as Xpert MTB/RIF Ultra. This upgrade notably improved the diagnostic performance of the GeneXpert^®^ system, particularly in detecting TB in patients with low bacillary burden (paucibacillary patients) [18,52].

The upgraded Xpert MTB/RIF Ultra cartridge enhanced case detection, especially for smear-negative, HIV-positive patients. However, its reduced specificity, particularly in previously treated patients, presents challenges. While the WHO recommends immediate treatment for Ultra-positive HIV patients, retesting is advised for others. Challenges arise from the test’s inability to determine rifampicin resistance in trace samples and the increased number of inconclusive results. Implementing Ultra at lower-level health facilities necessitates simple protocols and thorough training. Further investigation is needed to distinguish patients requiring treatment from those needing further examination, considering potential complications. Overcoming resource limitations remains pivotal for successful implementation, demanding enhanced access to culture and drug susceptibility testing. Provisioning ample cartridges is crucial to prevent shortages. Overall, further research is essential for the optimal interpretation and effective integration of Ultra in high-burden TB regions [53].

The pilot study for the implementation of the GeneXpert^®^ system in Brazil [15] permitted us to derive insights to enhance TB diagnosis: (i) Municipalities should guarantee training to technicians of the laboratories of the public service network so that they feel safe when using the new technology presented; (ii) for the expansion of the use of GeneXpert^®^ system in the country, a local provider is needed to maintain the GeneXpert^®^ system in addition to making cartridges always available; (iii) sputum smear microscopy should be maintained for follow-ups or sputum samples scarce, as well as for initial diagnosis of relapse/retreatment cases; (iv) the medical information system needs constant updates for the efficient use of the equipment.

A novel Xpert MTB/XDR test endorsed by the WHO and launched in 2020 can also detect resistance to isoniazid, fluoroquinolones, ethionamide, and amikacin [54,55]. A recent study involved two multicenter trials from high MDR/RR-TB burden countries, with 1228 participants for TB detection and 1141 for DR detection. The Xpert MTB/XDR showed high sensitivity (98.3% to 98.9%) and varying specificity (22.5% to 100.0%) for pulmonary tuberculosis. Sensitivities for isoniazid (94.2%), fluoroquinolone (93.2%), ethionamide (98.0%), and amikacin (86.1%) resistance were determined, as well as specificities ranging from 98.0% to 99.7%. False positives ranged from 1% to 29%, while false negatives were between 0% and 8% for the various types of resistance [55].

It is noteworthy that the newly introduced GeneXpert device, equipped with 10-color optics, was designed for use in conjunction with the Xpert MTB/XDR. However, it is crucial to highlight that the Xpert MTB/XDR cannot be operated on the existing 6-color instruments [54,55]. This implies that acquiring new instruments and providing retraining is imperative to prevent any potential complications.

In our research, we observed an important issue in places equipped with GeneXpert^®^ systems: the failure to transmit results due to the absence of an automated process for sending GeneXpert^®^ system reports to the Laboratory Environment Management System (Gerenciador de Ambiente Laboratorial—GAL). As a solution, we recommend the development of an interface between these two systems to address the problem of incomplete cartridge consumption data reporting.

It is important to highlight that the reduced levels of TB cases observed in 2020 were a consequence of the pandemic’s impact, rather than the GeneXpert^®^ assay serving as a preventative measure for TB. During 2020 and 2021, the global TB incidence was significantly influenced by the COVID-19 pandemic, as reported by the WHO. This influence was primarily attributed to the disruptions in essential TB services, leading to a decline in the diagnosis and official reporting of new TB cases in many countries [56].

Despite the limitations posed by discrepancies in database reports, attributed to the constantly evolving system, our study pointed out a current weakness in that there is an uneven distribution of RMT laboratories among regions and federal units, with a greater concentration in the Southeast region and specifically in the state of São Paulo, owing to its larger population. This disparity has implications for the detection of new TB cases in other regions, as we observed a direct correlation between the number of RMT laboratories and the frequency of MTB detection by the RMT in our analysis.

Another aspect for improvement rests on the fact that the majority of RMT laboratories are situated in capital cities, metropolitan areas, or reference centers, highlighting the need for greater dissemination of this equipment, especially in federal units where TB incidence is high or very high and the existing RMT laboratories are insufficient to meet the territorial demand. This challenge may arise due to the limitations in both physical and human resources available for conducting these tests and the complexities involved in organizing the sample collection and transport network, which often requires bipartite agreements.

The progress in managing the TB burden and its control exhibits notable variations among different regions and states in Brazil. Achieving the global targets established by the WHO within the framework of the End TB Strategy remains a complex challenge. Therefore, meeting the objectives outlined by the Brazilian Ministry of Health will necessitate substantial and comprehensive changes in healthcare accessibility, coverage, and quality. Consequently, it is imperative to direct efforts towards strengthening TB surveillance and control measures in various regions and states across Brazil, emphasizing early diagnosis of active cases and the initiation of appropriate treatment.

## 5. Conclusions

This study documented the introduction and utilization of the RMT-TB GeneXpert^®^ system (Xpert MTB/RIF and Xpert MTB/RIF Ultra) for TB diagnosis in Brazil. An assessment of the national distribution of laboratories equipped with these technologies and their capacity for TB and RIF resistance detection was conducted from 2014 to 2020 across all regions and federal units of the country. The findings revealed an uneven allocation of RMT laboratories across various Brazilian regions and states, with the Southeast region and the state of São Paulo accounting for approximately 39.4% to 45.9% and 20.2% to 34.1% of these facilities, respectively. These observations prompted recommendations for optimizing the distribution of RMT-TB resources and enhancing TB surveillance efforts in areas with higher incidence rates throughout the nation. This research expanded our understanding and bolstered the evidence base concerning the geographical distribution of rapid molecular testing to facilitate swifter diagnosis and improved disease control.

## Figures and Tables

**Figure 1 tropicalmed-08-00483-f001:**
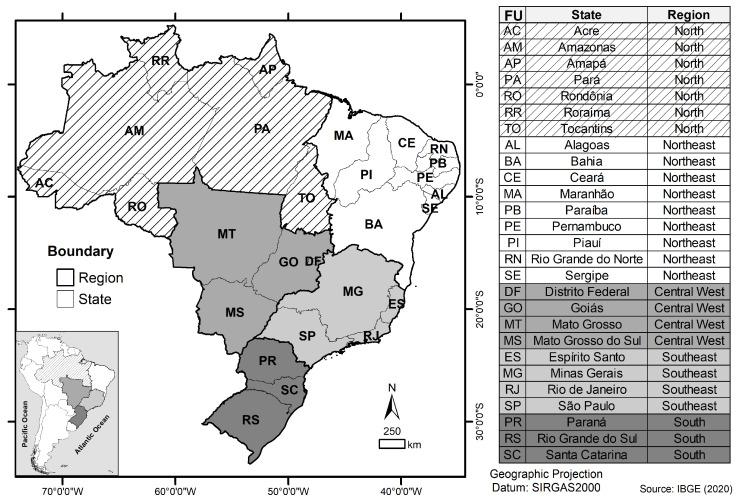
Stratification of Brazil’s geographic regions and federal units (states).

**Figure 2 tropicalmed-08-00483-f002:**
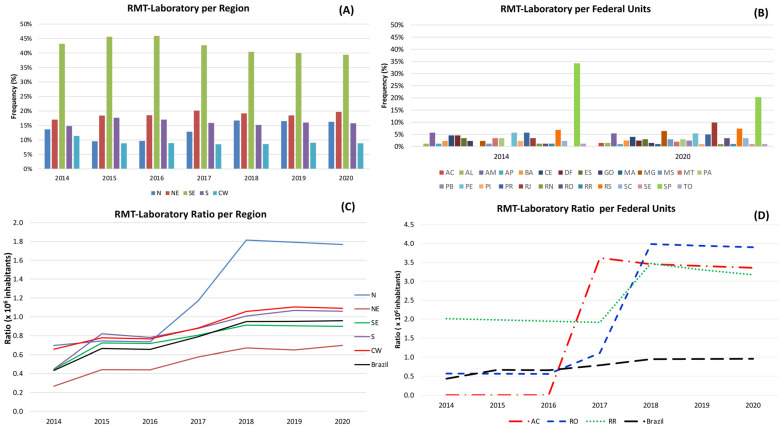
Frequency of laboratories conducting the rapid molecular test (RMT) using the Gene Xpert^®^ system in Brazil as well as the ratio of RMT laboratories to the population, both categorized by federal units (states) and regions. (**A**) Number of RMT per laboratory per region (N: north, NE: northeast, SE: southeast, S: south, and CW: central-west). (**B**) Number of RMT per laboratory per federal unit (FU) which are: Acre (AC), Alagoas (AL), Amazonas (AM), Amapá (AP), Bahia (BA), Ceará (CE), Distrito Federal (DF), Espírito Santo (ES), Goiás (GO), Maranhão (MA), Minas Gerais (MG), Mato Grosso do Sul (MS), Mato Grosso (MT), Pará (PA), Paraíba (PB), Pernambuco (PE), Piauí (PI), Paraná (PR), Rio de Janeiro (RJ), Rio Grande do Norte (RN), Rondônia (RO), Roraima (RR), Rio Grande do Sul (RS), Santa Catarina (SC), Sergipe (SE), São Paulo (SP), and Tocantins (TO). (**C**) Ratio of RMT per laboratory per region distributed by year from 2014 to 2020. (**D**) Ratio of RMT per laboratory per three states from the N region: AC, RO, and RR.

**Figure 3 tropicalmed-08-00483-f003:**
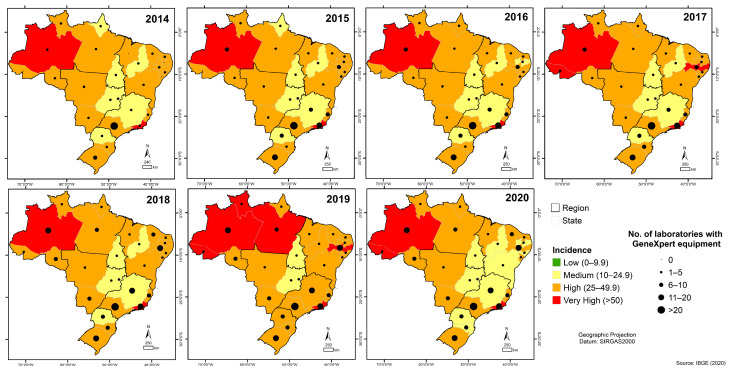
Spatial representation of laboratories conducting the rapid molecular test using the Gene Xpert^®^ system in Brazil in relation to the distribution of tuberculosis incidence from 2014 to 2020.

**Figure 4 tropicalmed-08-00483-f004:**
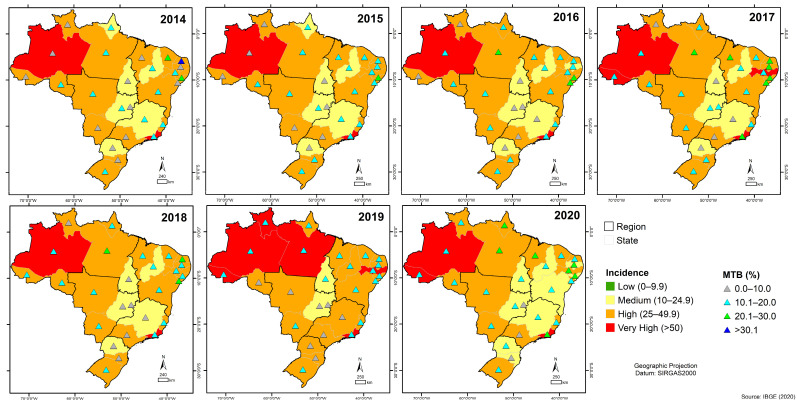
The geographical distribution of tuberculosis cases detected by the rapid molecular test (Gene Xpert^®^ system) in Brazil corresponding to tuberculosis incidence levels from 2014 to 2020.

**Figure 5 tropicalmed-08-00483-f005:**
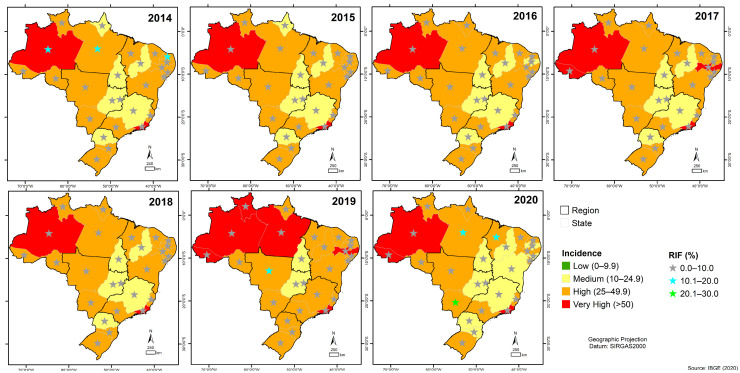
The geographic distribution of cases of *Mycobacterium tuberculosis* rifampicin resistance identified through rapid molecular test equipment, in conjunction with the distribution of tuberculosis incidence rates across Brazil from 2014 to 2020.

## Data Availability

All relevant data is presented within the manuscript.

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
