# Peer review of "The Role of GeneXpert® for Tuberculosis Diagnostics in Brazil: An Examination from a Historical and Epidemiological Perspective"

_tropicalmed, 2023, doi:10.3390/tropicalmed8110483_

Round 1
Reviewer 1 Report
TB is still a significant global health concern, and the incidence of TB shows a slowing increasing especially in Brazil. GeneXpert® is a useful rapid molecular test for TB detecting, as well as the RIF resistance identifying. The utilization of GeneXpert® assay in Brazil from both a historical and epidemiological standpoint would be usefully for the policy-making in TB prevention. However, this manuscript might be accepted after minor revision.
1. There is a significant increase in the tuberculosis incidence at 2019, such as in the state of PR, MG, PI. However, the tuberculosis incidence at 2020 in these states reduced to the level of 2018. It is interesting. Does the GeneXpert® assay benefit this TB prevention?
2. No Figure caption for Figure 2(A)-2(D).
Minor editing of English language required.
Author Response
Comments and Suggestions for Authors
TB is still a significant global health concern, and the incidence of TB shows a slowing increasing especially in Brazil. GeneXpert® is a useful rapid molecular test for TB detecting, as well as the RIF resistance identifying. The utilization of GeneXpert® assay in Brazil from both a historical and epidemiological standpoint would be usefully for the policy-making in TB prevention. However, this manuscript might be accepted after minor revision.
1. There is a significant increase in the tuberculosis incidence at 2019, such as in the state of PR, MG, PI. However, the tuberculosis incidence at 2020 in these states reduced to the level of 2018. It is interesting. Does the GeneXpert® assay benefit this TB prevention?
Authors’ reply: Your emphasis on this aspect is much appreciated. We have included the following discussion to ensure the accurate interpretation of such data within the paper: “It is important to highlight that the reduced levels of TB cases observed in 2020 were a consequence of the pandemic's impact, rather than the GeneXpert® assay serving as a preventative measure for TB. During 2020 and 2021, the global TB incidence was significantly influenced by the COVID-19 pandemic, as reported by the WHO. This influence was primarily attributed to the disruptions in essential TB services, leading to a decline in the diagnosis and official reporting of new TB cases in many countries [56].” (lines: 476 to 481).
2. No Figure caption for Figure 2(A)-2(D).
Authors’ reply: Correction performed.
General authors' comment: As you pointed out we could improve the results and conclusions for more clarity, we noticed that the introductory section of our discussion was wrongly deleted. Thus, we have enhanced the entirety of the discussion, fostering greater connections between the results and conclusions. Thank you so much for this input!
Reviewer 2 Report
I propose to change the title (it is not relevant to the content) to make it clear that the issue of the paper is tuberculosis (the word tuberculosis does not appear in the title). Furthermore, the authors do not evaluate the characteristics of the assays performed in the GeneXpert system but their role and importance in diagnosing tuberculosis.
In the abstract: I suggest replacing the initial phrase in the first sentence of the abstract to “The rapid molecular tests performed on the GeneXpert® system” and so on or provide the exact names of the tests used in Brazil for TB diagnosis (Xpert MTB/RIF, Xpert MTB/RIF Ultra).
In the 6th line of the abstract- instead of “...and TB and RIF detection by RMT should be “ ...and TB and resistance to RIF detection”.
In the introduction, the name of the manufacturer is given twice (Cepheid in Sunnyvale, USA) while once is enough. The same on page 3.
The paper is very elaborately written, demonstrating the positive role of the deployment of rapid molecular tests for improving the identification of TB cases.
Author Response
Comments and Suggestions for Authors
I propose to change the title (it is not relevant to the content) to make it clear that the issue of the paper is tuberculosis (the word tuberculosis does not appear in the title). Furthermore, the authors do not evaluate the characteristics of the assays performed in the GeneXpert system but their role and importance in diagnosing tuberculosis.
Authors’ reply: We appreciate your feedback on this important aspect. We changed the title from “Assessing GeneXpert® in Brazil: An Examination from a Historical and Epidemiological Perspective” to “The Role of GeneXpert® for Tuberculosis Diagnostics in Brazil: An Examination from a Historical and Epidemiological Perspective”.
In the abstract: I suggest replacing the initial phrase in the first sentence of the abstract to “The rapid molecular tests performed on the GeneXpert® system” and so on or provide the exact names of the tests used in Brazil for TB diagnosis (Xpert MTB/RIF, Xpert MTB/RIF Ultra).
Authors’ reply: Indeed, the best approach should be the “GeneXpert® system” as we evaluate the context of both versions (Xpert MTB/RIF, Xpert MTB/RIF Ultra). For consistency, we also corrected the entire text.
In the 6th line of the abstract- instead of “...and TB and RIF detection by RMT should be “ ...and TB and resistance to RIF detection”.
Authors’ reply: Correction performed.
In the introduction, the name of the manufacturer is given twice (Cepheid in Sunnyvale, USA) while once is enough. The same on page 3.
Authors’ reply: Correction performed.
The paper is very elaborately written, demonstrating the positive role of the deployment of rapid molecular tests for improving the identification of TB cases.
General author's comment: During this review, we noticed that the introductory section of our discussion was wrongly removed during the last edition before submission. Thus, we have enhanced the entirety of the discussion, fostering greater connections between the results and conclusions. Thank you for your input!
Reviewer 3 Report
Congratulations to the authors for presenting an interesting and important description of the implementation of rapid molecular testing for TB in Brazil. It is always important to consider program implementation and the relative success or ongoing challenges in historical context.
A further strength is the combination of data from policy repositories and health service implementation to provide a rich narrative overlaid onto geospatial representations. These are coherent and well-organized.
Two challenges arise: (1) the number of TB cases identified will increase with the implementation of any successful diagnostic tool, and GeneXpert has demonstrated effectiveness. Similarly, rational deployment of limited resources like GeneXpert means health authorities must the GeneXpert to places where there is suspected higher need / expected good yield. We therefore have correlation between GeneXpert use and incident TB diagnosis ratios that are fundamentally complicated. The authors are requested to engage this limitation directly, especially with regard to recommendations such as lines 355-358.
Relatedly (2) are lines 364-372 recommendations premised on the findings presented in this manuscript, or those from a previous (unpublished / uncited?) analysis of the pilot? The inferences / recommendations are not supported by the findings presented. It seems odd that they sort of 'arrive from nowhere' in the text. Overall, this seems to show an inherent tension between the findings that are largely descriptive and an impulse to draw more declarative conclusions.
Finally, it was strange not to have any discussion section in which the findings from this analysis are contextualized relative to other literature at all. Similarly, that the strengths and weaknesses may be stated directly. And recommendations for future research / policy / practice. Stylistically, the flow from descriptive findings directly into a sub-section on conclusion is unsatisfying.
A few sections where there are minor typos or the writing could be clearer. Only a final proof reading needed, no concerns. E.g., line 364 "The pilot study for the implementation of GeneXpert in Brazil [15] allowed draw lessons for the good diagnosis of TB" seems to have a missing word between allowed and draw?
Author Response
Comments and Suggestions for Authors
Congratulations to the authors for presenting an interesting and important description of the implementation of rapid molecular testing for TB in Brazil. It is always important to consider program implementation and the relative success or ongoing challenges in historical context. A further strength is the combination of data from policy repositories and health service implementation to provide a rich narrative overlaid onto geospatial representations. These are coherent and well-organized.
Two challenges arise:
(1) the number of TB cases identified will increase with the implementation of any successful diagnostic tool, and GeneXpert has demonstrated effectiveness. Similarly, rational deployment of limited resources like GeneXpert means health authorities must the GeneXpert to places where there is suspected higher need / expected good yield. We therefore have correlation between GeneXpert use and incident TB diagnosis ratios that are fundamentally complicated. The authors are requested to engage this limitation directly, especially with regard to recommendations such as lines 355-358.
Authors’ reply: Your observation regarding this underexplored finding in our report and subsequent discussion is much appreciated. This oversight occurred during the translation process from Portuguese to English, resulting in the loss of the data's intended meaning and a less effective representation within the discussion. We observed that the section heading of section “4. Discussion” was deleted. You can now find our enhanced treatment of this matter in the final paragraph of the results section.
Relatedly (2) are lines 364-372 recommendations premised on the findings presented in this manuscript, or those from a previous (unpublished / uncited?) analysis of the pilot? The inferences / recommendations are not supported by the findings presented. It seems odd that they sort of 'arrive from nowhere' in the text. Overall, this seems to show an inherent tension between the findings that are largely descriptive and an impulse to draw more declarative conclusions.
Authors’ reply: This matter is closely linked to the following issue, which corresponds to the first paragraph of the Discussion section. Nevertheless, to comprehensively address our findings, we have reworded this paragraph, incorporating additional context and references. We appreciate your valuable feedback on this regarding.
Finally, it was strange not to have any discussion section in which the findings from this analysis are contextualized relative to other literature at all. Similarly, that the strengths and weaknesses may be stated directly. And recommendations for future research / policy / practice. Stylistically, the flow from descriptive findings directly into a sub-section on conclusion is unsatisfying.
Authors’ reply: Indeed, this was an important problem to report. We did have a section “Discussion” starting from “The SUS expanded its network of laboratories equipped with RMT-TB technology”, however, it was wrongly removed from the many authors inputs. We added it back. We contextualized with a richer discussion, stating the strengths and weaknesses, as well as suggesting recommendations for future research/policy/practice.
Comments on the Quality of English Language
A few sections where there are minor typos or the writing could be clearer. Only a final proof reading needed, no concerns. E.g., line 364 "The pilot study for the implementation of GeneXpert in Brazil [15] allowed draw lessons for the good diagnosis of TB" seems to have a missing word between allowed and draw?
Authors’ reply: Thank you for highlighting this gap. We have addressed this paragraph (“permitted to derive insights to enhance TB diagnosis”) and conducted a thorough proofreading of the entire manuscript.
Round 2
Reviewer 3 Report
Thank you to the authors for the revisions. I understand that an error in formatting and content had crept into the previously submitted version. This resolved, the manuscript makes a much more coherent and valuable contribution.